# Preparation and Application of Metal Nanoparticals Elaborated Fiber Sensors

**DOI:** 10.3390/s20185155

**Published:** 2020-09-10

**Authors:** Jin Li, Haoru Wang, Zhi Li, Zhengcheng Su, Yue Zhu

**Affiliations:** 1College of Information Science and Engineering, Northeastern University, Shenyang 110819, China; 20173869@stu.neu.edu.cn (H.W.); 20173979@stu.neu.edu.cn (Z.L.); 20173914@stu.neu.edu.cn (Z.S.); 20174129@stu.neu.edu.cn (Y.Z.); 2State Key Laboratory of Applied Optics, Changchun Institute of Optics, Fine Mechanics and Physics, Chinese Academy of Sciences, Changchun 130033, China; 3Key Laboratory of Data Analytics and Optimization for Smart Industry (Northeastern University), Ministry of Education, Shenyang 110819, China

**Keywords:** LSPR, metal nano-particles, fiber sensors, plasmonics

## Abstract

In recent years, surface plasmon resonance devices (SPR, or named plamonics) have attracted much more attention because of their great prospects in breaking through the optical diffraction limit and developing new photons and sensing devices. At the same time, the combination of SPR and optical fiber promotes the development of the compact micro-probes with high-performance and the integration of fiber and planar waveguide. Different from the long-range SPR of planar metal nano-films, the local-SPR (LSPR) effect can be excited by incident light on the surface of nano-scaled metal particles, resulting in local enhanced light field, i.e., optical hot spot. Metal nano-particles-modified optical fiber LSPR sensor has high sensitivity and compact structure, which can realize the real-time monitoring of physical parameters, environmental parameters (temperature, humidity), and biochemical molecules (pH value, gas-liquid concentration, protein molecules, viruses). In this paper, both fabrication and application of the metal nano-particles modified optical fiber LSPR sensor probe are reviewed, and its future development is predicted.

## 1. Introduction

Surface plasmon resonance (SPR) refers to a phenomenon when photons cause the oscillation of the free electron on the metal surface when light is incident on the metal-dielectric interface, and the wave vector component of a light wave along the metal interface matches the wave vector of SPR [1,2]. In the oscillation process, the energy of a light wave is converted into the oscillation energy of the free electron, causing the strong attenuation of light field intensity, where the obvious absorption peak appears at the resonance wavelength position in reflection or transmission spectrum [3]. SPR is divided into long-range SPR (LR-SPR) and localized SPR (LSPR) according to the transferring distance or light field distribution of SPR wave (SPRW) [4,5]. LR-SPR exists in metal micro/nanostructures, including two-dimensional, three-dimensional structures, and the surface of metal micro/nanowires [6]. This type of SPRW usually propagates along a specific direction and has a certain linear transmission length along the interface. It can be used in all-optical signal modulation, imaging, and biochemical sensors. LSPR exists on the surface of metal nano-particles (NPs) with various morphologies, which can be spherical, ellipsoidal, or even random shape [7,8]. This type of SPRW is limited on the surface of metal NPs, forming an evanescent field around it, so it is very sensitive to the surrounding environment [9,10], mainly being used in the detection and screening of biomolecules and living cells, biochemical reaction process monitoring, biological surface analysis and treatment [11,12,13,14].

In recent years, to explore the optical fiber SPR sensors, the sensing part is modified by the micro/nanostructure of the optical fiber itself, such as using a metal film with nano-thickness or metal nano-array as the sensing layer to replace the coating layer of the optical fiber [15,16]. Or the end face of the optical fiber probe is modified by micro/nanostructure, coated by nano-size noble metal layer or etched by metal nanostructure as the sensing part of the SPR optical fiber sensor [17,18]. These new structures benefit from the progress of metal coating technology and micro/nanostructure processing technology in chemical or physical fields and are the result of interdisciplinary and mutually promoting development [19,20]. In the sensing measurement, the sensing part of the optical fiber will be affected by the object to be measured, so the sensing purpose can be realized by detecting the change of transmitted light or reflected light signal in the optical fiber [21,22]. The research on the high field localization of LSPR generated by metal micro/nanostructures such as single metal NP, NP pairs, and metal NPs arrays to effectively improve the sensitivity, selectivity, spatial resolution, and integrability of SPR sensors and all-optical devices, which has become an important research and application direction in the field of SPR technology in recent years [23,24,25,26,27].

Using “Plasmon,” “Plasmon nanoparticles,” and “Plasmon nanoparticles fiber” as the keywords in Google Scholar, the corresponding number of published papers was compared in Figure 1. Among the reported plasma related articles, 46% were related to NPs. The work on metal NPs modified fiber SPR has been reported since 2000, and almost all the articles (98.6%) have appeared in the last decade.

In this paper, the fabrication methods and technical characteristics of LSPR fiber structures are compared and analyzed, including fiber Bragg grating (FBG), tilted fiber Bragg grating (TFBG), long-period fiber grating (FPG), D-typed fiber, U-shaped fiber, micro/nano-fiber, photonic crystal fiber (PCF), cascaded fiber and fiber end face (usually represented by Fabry-Perot interference). Metal NPs were modified on the surface of the optical fiber structure by polymer sol dispersion and dip coating, electrostatic adsorption layer by layer self-assembly, nano-lithography, and molecular capture. Furthermore, the work in the field of biochemical sensing is carefully compared and summarized, covering the application in biological cells, chemical molecules, heavy metal ions, pH value, and gas.

## 2. Common Optical Fiber Structure and Probe Fabrication Method

The excitation conditions of LSPR effect on the surface of metal NPs include: specific light field transmission direction and strong enough light field. Optical fiber provides an ideal optical transmission path in which the optical signal is efficiently transmitted without being affected by the large angle bending from the fiber in space. Different fiber structures can provide the required light intensity for the excitation of LSPR effect by guiding the optical signal into the functional film with the help of the evanescent field, optical reflection, and scattering. In recent years, the frequently reported fiber structures are shown in Figure 2, including gratings (FBG, TFBG, FPG), micro-configurations (D-typed fiber, U-shaped fiber, micro/nano-fiber), composite structures (cascaded fiber, doped fiber), and reflection ends (fiber flat end, fiber taper, Fabry–Perot interferometer).

Among them, the working mechanism of grating structures is to establish the interaction relationship between the optical signal and the measurement parameters, where, the sensitive material has elastic deformation function under the action of the measured variable; the micro-configurations and composite structures separate high-order modes and low-order modes to produce the multimode interference effect. In these structures, the equivalent refractive index of the sensitive material coating will have an effect on the optical phase of the higher-order mode; the reflection ends can directly change the scattering and stimulated characteristics of the optical signal passing through the functional materials on the end surface of optical fibers.

The simplest LSPR-fiber probe can be obtained by directly coating polymer sol doped with metal NPs on the end face of optical fiber [28]. The sol coating layer has been used as the interference cavity of the Fabry–Perot structure, in which the metal NPs are uniformly dispersed. Although this kind of probe has been verified to measure ethanol concentration, more contribution may come from the gas adsorption of polymer sol. Moreover, metal NPs are difficult to interact with the outside environment, and their random distribution will also affect the quality of interference light. Liu et al. prepared a shape controllable metal NPs elaborated plasmon probe at the end face of the optical fiber, as shown in Figure 3a, and realized the non-contact determination of IgG concentration combining with microfluidic technology. The corresponding production cost is low, but the process is complex, including polymer sol ball self-assembly, soap film coating, violet lithography, polymer removal, magnetron sputtering, etc. [29]. In general, ordered metal NP arrays can be prepared by fewer procedures mentioned above. The polymer NPs spherical sol forms a single-layer crystal structure on the plane structure and is transferred onto the fiber end face [30], as shown in Figure 3b. N. Polley, et al. suggested another interesting technique by transferring the Au NPs film from a flat glass substrate to the optical fiber tip [31] where the gold film was peeled off in NaOH solution and bund on fiber tip via the amine groups of (3-aminopropyl) triethoxysilane. 

Polymer gel is an ideal carrier and dispersant for solid NPs. It provides a simple way to make metal NPs functional membranes [32]. The material, size, morphology, polydispersity index, wavelength, and environmental refractive index of NPs affect the absorption and scattering results of incident light, which are analyzed by Mie scattering theory. In addition to their own morphology, due to their size distribution in several to dozens of nanometers, the arrangement of NPs in the functional film deposited on the optical fiber will determine its optical and information interaction performance [33]. The layer by layer self-assembly method provides a simple, flexible, and efficient way to accurately control the thickness and structure of NPs functional films. Various surface-treated NPs (metals, metal oxides, polymers, luminescent materials, or biomolecules) were used to construct multilayer structures by electrostatic attraction.

Similar multilayer structures can also be achieved by in-situ synthesis. The difference lies in whether the NPs are modified on pre-deposited multilayer polymer films. With the help of the photochemical deposition method and controlling the laser irradiation time, Ag NPs were rapidly deposited on the end face of the fiber, which was inserted into the reaction solution for measurement [34]. This provides a simple and efficient method for the deposition of NPs on the fiber end face. However, the influence of the random morphology of silver NPs on the reflected light and sensing characteristics of fiber probe is worthy to be studied further. The particle size of Au NPs in sol affects its SPR absorption spectrum, which usually shows a wide absorption spectrum (several nanometers to tens of nanometers) similar to the fluorescence spectrum [35]. Spasopoulos et al. dip-coated Au NPs sol on the U-shaped optical fiber and irradiated it with high-energy laser pulse (532 nm, 6–7 ns duration, 6–10 mw/cm^2^) to change the cluster morphology, resulting in the photothermal dissociation and the sharp plasma resonance peak [36], the corresponding optical fiber probe is shown in Figure 3c. Laser pulse photothermal treatment can precisely adjust the morphology and size of Au NPs by setting laser wavelength, duration, intensity, pulse width, etc. in sol and liquid environment [37]. This method has been verified and has been used in many works, and even has been used to repair biological tissues [38].

In addition to polymer dispersion, the self-assembly of metal NPs has been realized by simple chemical adsorption. Before that, the silica surface of optical fiber needs to be silanized or amino-modified [39]. Furthermore, Ag NPs were loaded with other types of metal NPs to obtain better sensing properties [40]. For PCFs with special structures, the SPR film was prepared when the solution of Au NPs flows slowly through the inner wall of the modified hollow core, as shown in Figure 3d. The NPs density depends on the volume ratio of bonded and unbound silanes [41]. 

As reviewed in Hu’s work [42], the NPs functional film on the inner wall of the PCF air hole have been demonstrated to be prepared by amino modification, stack and draw technology, high-pressure chemical deposition technology, dynamic low-pressure chemical deposition technology, etc. The SPR effect of metal NPs can improve their optical and sensing properties with the help of graphene two-dimensional materials [43]. The upper and lower layers of graphene oxide (GO) films and Ag NPs are fabricated by dip coating and photochemical deposition process, respectively, so as to obtain GO/AgNPs/GO sandwich film on the end face of optical fiber. It should be pointed out that the probe must be kept in a nitrogen environment before being used to prevent the oxidation of Ag NPs. In addition, the flexible polymer materials doped with metal NPs are made into stretch sensor devices called electronic skin, whose stability and reliability becomes a huge obstacle to its commercial application [44]. Based on metal NPs, flexible polymers, and SPR effect, the stretchable flexible optical fiber sensor is expected to become a new research direction and be used for exploring bionic skin, flexible clothing, and all-optical monitoring function layer. These pave the potential application way to the high-performance sensors for a harsh environment such as high electromagnetic interference, high temperature, and high humidity.

## 3. Fiber-LSPR Refractive Index Sensors

The structures and modification materials of different fiber-LSPR refractive index sensors, as well as their sensitivity and operating range are compared in Table 1.

Jiang has put forward a relatively perfect design scheme of metal NPs optical fiber SPR sensing probe very early [45], which was improved and optimized in many similar works. The fabrication process is indicated in Figure 4.

In this classical structure, the LSPR of Ag NPs is further enhanced by the SPR effect of Ag nano-film, which is uniformly dispersed and fixed on the fiber surface by polyvinyl alcohol (PVA) and covered by graphene to improve the energy transfer efficiency and delay the oxidation process of Ag NPs. Laser-induced deposition and dip-coating are the most effective methods for elaborating the metal NPs onto the fiber surface [46]. SPR spectrum of metal NPs was timely monitored and became saturate after a proper deposition time. Both the position and intensity of the absorption peak are determined by the resonance condition of SPR and the optical loss was caused by the change of refractive index difference. The LSPR effect of metal NPs can be excited by the optical evanescent field, and the free electrons in the metal are launched onto their surface, thus providing the possibility of self-assembly of metal NPs (similar to electrostatic adsorption self-assembly) [47]. The evanescent wave assisted coating technology has been used to deposit thick metal NPs function film on micro/nano-fibers. However, there is no chemical hinge among the NPs on the film surface, resulting in its loose structure, which is easily destroyed when it is used as a liquid refractive index sensor.

Double layered polydopamine (PDA) sandwich Ag NPs was proposed, where the first/inner layer served as the substrate with strong adhesion and reductant to improve the deposition process of Ag NPs, the second/outer layer covered on the Ag NPs to prevent them being oxidation and increase the sensitivity [48]. The disadvantage is that the multi-layer film leads to the wide LSPR spectrum (~ 300 nm), which is disadvantageous to improve the resolution of the sensor. Moreover, the accumulation of the layers, thickness, and composition of multilayer electrostatic assembled functional films stimulated the new loss mode resonance (LMR), which may also occur when the film is thin enough, but it will not affect the intensity and distribution of SPR on the spectrum [49]. This LMR based muti-layers film probe was experimentally demonstrated with the sensitivity of 11.2 nm/% RH during the range of 45–90% RH.

Metal NPs dispersed in the liquid was also used to measure the change of refractive index, but the experimental results indicate that the sensitivity has been significantly improved by modifying the surface of optical fiber [50] (from 475 nm/RIU to 892 nm/RIU). In addition, due to the strong evanescent field around the micro/nano-fiber, the SPR effect of more metal NPs was excited to obtain a higher sensitivity. The metal NPs on the surface of the LSPR optical fiber sensor may be destroyed and washed by liquid samples frequently. Mos et al. encapsulated micro/nano-optical fiber in Au NPs liquid crystal suspension to measure the environmental temperature, electric field, and magnetic field, by using the refractive index characteristics of liquid crystal under different external environment [51]. The refractive index sensing structure based on single Ag nano-rod and side-polished suspend-core microstructured optical fiber (SPSC-MOF) was theoretically designed and demonstrated with the sensitivity up to 8600 nm/RIU [52]. The effects of Au, Ag, Cu nanorods, and MOF structural parameters on the sensing properties have also been studied in detail. However, the fabrication of this structure is extremely difficult due to its extremely small size and fine structure. The structure of cascaded optical fibers with different core diameters is stable and compact, but the corresponding sensitivity is low [53]. Bandyopadhyay et al. obtained the ultra-high refractive index sensitivity near the inflection point of long-period fiber gratings [54]. The bandwidth gap of the double cladding mode of long-period fiber grating is finely adjusted to obtain the ideal phase matching spectrum curve. The functional film can also be used to modulate it by observing the transmission spectrum during the deposition of Au NPs.

Some simulation results show that the sensitivity of spherical Au NPs modification is much higher than that of gold nano-cubes and nano-triangles. This may be due to the difference in optical fiber structures and excitation angle of incident light [55]. The influence of metal NPs morphology on the performance of the optical fiber SPR sensor needs special attention and in-depth analysis in future research. Triangle shaped Ag NPs have been experimentally verified to have a higher refractive index sensing sensitivity of more than 3 times than that of spherical Ag NPs [56]. Moreover, GO coated on the surface has little effect on its sensing performance, but it can effectively prevent the oxidation of Ag NPs and fix them on the surface of optical fiber more firmly. The multi-branched sharp edges on Au NPs have been used as the hotspots to enhance the SPR intensity effectively [57]. 2–3 times enhancement of refractive index sensing was experimentally demonstrated in the different wavelength ranges. However, the contribution of multi-branch edges was failed to obtain by comparing its refractive index sensing performance with that of spherical Au NPs. In several works, the significant impact from the LSPR of metal NPs with spherical or other free shaped morphology on the SPR effect of pure metal film has been verified. Spasopoulos et al. have experimentally verified that the optical fiber refractive index sensor is constructed by combining dip-coating and tunable diode laser irradiation (TDLI) technology and observing the extinction spectrum in real-time, without strictly controlling the particle size uniformity of metal NPs [58]. During this process, TDLI technology has been used to regulate the size of metal NPs [59].

The coverage ratio of metal NPs on the surface of optical fiber is not directly proportional to the performance of the sensor. Wu et al. concluded through experiments and simulation analysis that when the surface coverage ratio of Au NPs is 15.2%, the refractive index probe based on coated muti-mode fiber (MMF) has the highest (saturated) sensitivity [60]. The specific value of the surface coverage ratio will vary for different fiber structures and metal NPs.

A theoretical simulation shows that the electric field strength of the composite film containing Au NPs is increased by 4–10 times, compared with the pure Au nano-film [61]. The enhancement factor is inversely proportional to the size of Au NPs. The coupling between the metal NPs and the metal film can enhance the electric field. The coupling resonance effect among different metal particles, metal films, and functional materials (graphene) uniformly causes the shift or even broadening of SPR resonance peak [62]. The former is mainly due to the difference of dielectric constant of materials, resulting in a different refractive index; the latter is caused by the energy transfer between different materials, resulting in electromagnetic dispersion. The experimental results show that the refractive index sensing sensitivity of muti-walls carbon nanotube (MWCNT)/Pt NPs composite structure modified fiber is 5923 nm/RIU, which is significantly higher than that of Au nano-film and MWCNT-Au composite film with SPR effect (the corresponding refractive index sensitivity are 1863 nm/RIU and 2524 nm/RIU, respectively) [63]. This work also found that the sensitivity of the proposed fiber probe will increase with the doping amount of Pt NPs, and the sensor has a better figure of merit.

## 4. Fiber-LSPR Chemical Sensors

The chemical sensors can detect the chemical molecules in solution, pH value, heavy metal ions and gas concentration. Both the advantages and disadvantages of different sensors will be compared in the following sections. 

### 4.1. Chemical Molecules

The effective combination of the chemical molecules to be measured and optical fibers, as well we the corresponding Raman emission efficiency will affect the final detection effect, which is embodied in the performance parameters of the sensor. The stimulated luminescence characteristics (wavelength, frequency, and intensity) and the size of metal NPs (corresponding to stimulated fluorescence of different wavelengths) are all factors to be considered [64]. The enhancement of the optical signal and the interaction efficiency of chemical molecules are determined by the properties of metal NPs and functional films. Atomic layer deposition technology was introduced to fabricate the protective Al film with a thickness of ~1 nm to encapsulate the Au NPs coated fiber nanotaper [65]. The Raman intensity of the nano-thickness of Al film has been experimentally demonstrated with the increasing factor of ~2 times. The change of resonance wavelength or intensity of the LSPR spectrum gradually decreases and tends to be flat, indicating the effective working range, which is mainly limited by the binding sites of the molecules to be measured and metal NPs [66]. The directly growing WS2 film was simply coated on the surface of optical fiber to support the growth of Au NPs and be beneficial for the evanescent field enhancement on the surface of optical fiber [67].

A sharp fiber tip was prepared by immersing a fiber end into the HF solution and being etched for 4 h [68]. It was later modified by Ag NPs and PH sensitive molecules (4-mercaptopyridine, 4-Mpy) using a laser-induced deposition technique. The reflected surface-enhanced Raman (SER) spectra were recorded to determine the PH values in solution with the sensitivity of 0–1 cm^−1^ per PH unit depending on Raman peaks and monitoring the PH change in cancer cell lines with a relatively large difference. Islam et al. adopt poly ethaline glycol (PEG) sol-gel to encapsulate Au nano-dendrities and phenolphthalein (PHPH) [69]. The mixture was later elaborated on the side surface of a non-core fiber (NCF) with a diameter of 1012 μm and indicated the fast response (0.87 s) to PH. 

The fiber structures also exert impact on the Raman intensity, which is used to achieve ultra-low detection limit for R6G concentration, as the Ag NPs elaborated U-shaped fiber tip proposed by Yin et al. [70]. The limit of detection (LOD) of 10 nM was experimentally demonstrated with the significant enhancement of Raman intensity for the U-shaped fibers with the length differs from 12–42 μm. Two separated fiber ends were used as the excitation and collection fibers by two V-grooves, respectively. The excitation efficiency can be adjusted by controlling the relative angle of the two V-grooves [71].

The combination of the surface evanescent field of the nanofiber and the LSPR of the metal NPs effectively improves the sensitivity of R6G by 1.06 × 10^6^ [72]. Biological electrospinning technology provides a simple and efficient method for polymer micro/nano-fiber and flexible doping of metal NPs. The active flow of the laser-heated air bubble suspended in the sol is adsorbed onto the fiber end surface to excite the LSPR effect [73]. The Raman enhancement factor of R6G is increased by 140 times compared with bare fiber. In addition to Au and Ag NPs, other metal NPs are also used to make LSPR fiber probes. Cu NPs can be used as the reduction catalyst of nitrate, which was converted into ammonia and adsorbed on the surface of carbon nanotubes (CNTs). Parveen et al. prepared the nitrate fiber probe according to this principle [74], it has the high sensitivity (80.62 × 10^6^ nm/M), selectivity (compared to carbon, sulfur, and iron), low detection limit (4 nm), and fast response time (15 s). 

### 4.2. Heavy Metal Ions

Heavy metal ions from industrial wastewater are easily enriched in organisms and seriously endangered human health. According to US Environment Protection Agency (USEPA) guidelines [75] and World Health Organization (WHO) standards [76], the indicators of Hg^2+^ ions in drinking water were <2 ppb (10 nm) and 6 ppb (30 nm), respectively, as Shukla et al. reported in their work [77]. Different functional materials have different working ranges, where, the glucose capped Ag NPs with low volume ratio (1:13) showed larger response for the Hg^2+^ solution with a lower concentration; but for the high concentration, this corresponding ratio was 1:27. The redox reaction of Hg^2+^/Hg and Ag+/Ag changed the size of glucose-Ag NPs and resulted in Ag-Hg amalgam formation. Hg^2+^ will become amalgam-like alloys after bonding on Au NPs due to the strong chemical affinity [78], which was experimentally demonstrated by dispersing different Hg^2+^ concentration into constant Au NPs solution and monitoring the color change. Poly (allylamine hydrochloride (PAH) and poly acrylic acid (PAA)-Au NPs were used as the polycation and polyanion, respectively, to assemble the bilayers function films of PAH/PA + AuNPs. Hg^2+^ probe was regenerated by immersing it into a HNO_3_ solution for 1 h. 

Polymer dispersing Au NPs and dip coating on the end face of optical fiber provides the simplest manufacturing method for heavy metal ion SPR fiber probe. In order to eliminate the interference of environmental factors and obtain the high sensitivity, the probe has been experimentally verified to have a minimum detection limit of 1 μm [79], which may be far higher than the stringent indicators of drinking water. Similarly, the polyelectrolyte, which can be used for electrostatic layer by layer self-assembly, includes poly(sodium-p-styrenesulfonate) (PSS, negatively charged layer) and poly-dimethyl diallyl ammonium chloride (PDDA, positive charged layer) [80]. With the number of layers, the refractive index difference between the core and cladding decreases, and the resonance peak gradually disappears. But the sensitivity increases to the maximum when the refractive index difference approaches zero. The sensitivity of the proposed Hg^2+^ probe was experimentally demonstrated with the high sensitivity of >0.1 nm/ppm, which is ~33.5 times compared with that of the uncoated and ~14.5 times compared with that of the PE-only coated LPFGs. Prakashan et al. proposed the non-toxic SiO_2_-TiO_2_-ZrO_2_ ternary system as the probable host for Cu@Ag core shell NPs to determine the Hg^2+^ concentration [81]. However, the contribution of SiO_2_-TiO_2_-ZrO_2_ ternary system is unclear in addition to its host role similar to the polymer so-gel. 

Different chemical groups or biomolecules are used as indicators of heavy metal ions, which can effectively improve the selectivity of the corresponding sensors. The -COOH group can interact with Pb^2+^ ions other than As^2+^, Cd^2+^, and Hg^2+^ to change the refractive index and dielectric constant of Au NPs, as well as the intensity of its SPR spectrum [82]. Many bioactive molecules are used as indicators of heavy metal ions. Fluorescence, electrochemistry, and colorimeter methods have been used to develop heavy metal ion sensors. “thymine-Hg^2+^-thymine base pair mismatches” mechanism was illustrated in a DNA-Au NPs based fiber sensor [83]. The improved LOD was contributed to both refractive index increase (around the Au NPs) and near field coupling enhancement (among Au NPs and DNA conjunction). Bovine serum albumin (BSA)-chitosan modified by Au NPs was coated on the surface of U-shaped optical fiber and was verified by experiments to measure Hg^2+^ in tap water, sewage, and soil with a LOD of 0.1 ppb, and the good selectivity [84]. A wild typed bacteria E. coli B40 being as the negatively charged cells were equipped on a PAH/PSS@Au NPs probe and used as the receptor of Hg^2+^ and Cd^2+^ [85]. Its selectivity for Hg^2+^ and Cd^2+^ is significantly higher than that of other ions (K^+^, Na^+^, Cu^2+^, Zn^2+^, Co^2+^, Ni^2+^, and Mn^2+^). 4-Mpy can not only be bonded to Au NPs closely but also selectively capture pyridine Hg^2+^ in water through nitrogen [86], as shown in Figure 5.

A multilayer composite structure (Au film/4-Mpy)-Hg^2+^-(Au NPs/4-Mpy) was formed on the surface of the optical fiber, where the Au NPs effectively enhanced the electric field strength of the Au film surface by more than 6 times. Moreover, the enhanced electric field confined between the Au NPs and the Au film is 2–3 times of that of the Au NPs, depending on the doping amount of the Au NPs.

Other sensing properties of the fiber optic probe for heavy metal ions have also been improved by other methods. SPR effect realizes the detection of heavy metal ions, and high selectivity has been proved to provide vacancy binding sites for ions by ion imprinting [87]. The concentration of various heavy metal ions can be measured on a single optical fiber probe at the same time. The Au NPs size, the 1, 1-Mercaptoundecanoic acid (MUA) drying temperature and the MUA concentration of each part were carefully controlled. The repeatability of the probe depends on the cleaning of heavy metal ions by the ethylenedinitrilotetraacetic acid (EDTA) on the surface of the contaminated probe [88]. A fiber-optic Pb^2+^ sensor was proposed and its hand-held system was designed to demodulate the wavelength position information using an FBG [89]. Although metal NPs and SPR effect are not introduced, it also provides a reference for the development of optical fiber sensor for heavy metal ions. 

### 4.3. Gas

Similar to the surface of optical fibers, the optical waveguides on planar substrates are more easily modified by functional materials. Ag NPs have higher activity and lower LOD than Au NPs, which can realize more precise monitoring of H2S concentration. Mironenko et al. verified that the LOD of the former is 0.1 ppm, and the difference between them is 50 times [90]. Subsequently, Au and Ag NPs modified U-shaped optical fiber VOCs gas probe was also proposed and verified [91]. However, it should be noted that the sensitivity of Au and Ag NPs to different VOCs is different. As Paul et al. reported in their work, the detection limit of Au NPs for propanol is ~3.81 ppm, compared to ~11.09 ppm for Ag NPs. They indicate a similar sensitivity to ethanol and the significant difference for methanol and acetone (LODAu: LODAg are 23 and 2.3, respectively).

Pt NPs modified microfiber cones and U-shaped optical fibers are respectively used to realize the sensing measurement of ammonia and hydrogen [92]. Gao et al. fabricated PMMA Pd NPs composite micro/nanofiber by doping metal NPs into polymer materials and adopting sol-step stretching method [93]. The optical fiber was then connected to the ordinary single-mode optical fiber, and the detection of hydrogen concentration below 1% with the sensitivity of 52 pm/ppm was realized.

Semiconductor nanomaterials are widely used in gas sensing technology, and their sensing sensitivity can be improved by modifying metal NPs on their surfaces. Chao et al. doubled the LOD of ethanol/acetone [94], and significantly improved the response time and recovery time. The introduction of metal NPs effectively reduces the working temperature of oxide semiconductor materials, which can realize the gas sensor at room temperature [95]. The performance of fiber-LSPR based chemical sensors reported in recent years is compared in Table 2. 

## 5. Optical Fiber LSPR Biosensors

High-performance fiber-optic LSPR biosensor probes are designed based on different fiber structures, as well as the modification of metal NPs, antibody, and marker. These biosensors have been used to detect glucose, protein, amino acid, and nucleic acid. The typical structures and characteristics of optical fiber probes are reported in this section in recent years. The role and optimization of metal NPs will be reviewed as well.

### 5.1. Typical Fiber Structures and Properties

It is well known that the initial SPR was realized by means of the reflection angle modulation of Otto and Krestschmann prism structures to satisfy the SPR coupled resonance condition. Flat surface optical fiber structures, such as rectangular and D-typed fiber, play a similar role [96]. The U-shaped fiber based on fiber microbend for light leakage, the micro/nanofiber with fine diameter fiber to excite evanescent field, the fiber gratings enhanced by multiple reflection interference, the microstructured fibers regulated by periodic array light, and the cascaded fiber with mode decomposition and interference provide the optical action surface to realize the excitation of SPR effect [97].

In order to obtain a stable U-shaped single-mode fiber, flame heating is used to further reduce the bending radius of the fiber. When pre-tension stress is applied to the single-mode fiber and fixed in the capillary [98]. The introduction of large diameter and flexible polymer optical fiber greatly reduces the manufacturing difficulty and cost of SPR optical fiber sensor, which can achieve the low precision measurement with low requirements for optical loss. The U-shaped fiber and etch plane are fabricated from PMMA fiber [99].

Micro/nanofiber provides a stronger optical field, called the evanescent field, to improve the intensity of the optical signal involved in the SPR effect, and enhance the interaction between the optical field and the environment to be measured [100]. In similar fiber structures, the penetration depth of the evanescent field and the optical scattering loss of NPs or structures need to be considered. If the biconical micro/nanofiber is stretched too thin (less than 4 μm) in the waist position, it is easy to be damaged in the cleaning and coating process of functional materials. The tail end of the single cone micro/nanofiber is flat cut by the fiber cutting pen and then coated with functional materials to make the reflective SPR probe [101]. Biconical micro/nanofibers are fabricated from ordinary single-mode fibers by high-temperature melt stretching. Oxy-hydrogen flame, electric heating, and plasma discharge provide a high-temperature environment of thousands of degrees Celsius, which is suitable for processing different types of optical micro/nanofibers [102]. Its lumbar diameter is precisely controlled by heating temperature and stretching speed, while the length of the cone area depends on the stretching speed and heating mode (scanning or single point fixed torch).

Polymer optical fiber was processed into a tapered optical fiber, which is easier to manufacture because of its low melting point and easy decomposition [59]. Tapered micro/nanofibers were obtained by dynamic chemical etching-drawing method [103]. The uncoated end of the single-mode fiber is immersed in hydrofluoric acid and is slowly lifted up to control the taper angle and taper length. The end face of the hollow-core fiber (HCF) is coated with Ag nanospheres and nanorods, and its micropores were easily obtained by capillary action [104]. The LOD of melamine was measured experimentally to be 100 nm, which was far lower than the minimum content of milk powder (1 mg/kg) and other foods (2.5 mg/kg).

For fiber structures with a single resonant peak, such as reflected light of FBG or transmission light of LPG, the ultra-high sensitivity can be obtained when the wavelength of the most resonant peak moves due to the sudden change of light intensity near the peak inflection point [105]. Optical signals can also be reflected to the sensing area through some special optical fiber structures and interact with the environment to be measured. TFBG has been used to effectively control the output direction of the optical signal, and its core mode has the adjustment function of temperature cross sensitive, as shown in Figure 6 [106].

Compared with the LRSPR sensor, the LOD and surface enhancement effects of the LSPR sensor are more obvious. Noble metal NPs provide a larger contact area and achieve efficient information exchange between the metal NPs and the analyte. More importantly, the NPs surface generates a higher concentration of electromagnetic field and the limitations of a single-point amplification effect, which can effectively enhance the detection efficiency of a biological analyte in nanometer size or spacing [107]. In the detection process, the excitation photons meeting the coupling frequency resonate with the surface electrons of the metal NP, thereby generating a stronger and concentrated surface-local evanescent field and realizing the single-molecule nanoprobe having a very low LOD [108].

The cascaded fiber structure guides the high mode light signal onto the fiber surface to improve the sensing performance. The multimode fiber (MMF) and PCF have been verified by experiments, where the LOD of biosensor has been increased to 1 nm, and the influence of environmental temperature was effectively eliminated to limitation less than 7.2 pm/°C [109]. Multiple fiber structures in series or in parallel can form the sensing arrays, which are expected to realize the simultaneous detection of multi-parameters or multi-component samples [110]. The air hole of PCF usually collapses when it is fused with MMF. In order to reduce the cost and improve the repeatability of the sensor probe, the HCF replaces PCF to construct the cascaded fiber structures [111]. The chemical etching method is widely used to fabricate the tapered micro/nanofibers from single-mode fiber, MMF, PCF, etc. [112]. With the decrease of the fiber diameter or the destruction of the cladding structure, the optical signal is leaked to the sensing area. The sensing signal was doubled by constructing multiple sensing regions, but the length of each sensing arm must be strictly controlled to be equal [113]. The different length of the interference region causes the multiple overlapping interference spectra for the multi-mode interference, Mach–Zenhder, Fabry–Perot structures, and other fiber interferometers. This reason results in the demodulation difficulty and failing to enhance the sensing effect.

The highly integrated optical fiber SPR sensor and planar structure can be fixed by a microfluidic channel [114]. At the same time, it is easier to replace the tested sample with less consumption, and the sample pool is easier to be cleaned. Focused ions beam (FIB) technology has been applied to prepare the Au NPs array at the end face of optical fiber and realized the precise control of the size and morphology of Au NPs [115]. However, high cost and technical difficulty will become a huge obstacle to the commercialization of such devices. In order to improve the surface uniformity of the fiber, the silica NPs were self-assembled on the fiber surface [116]. The sensitivity is affected by the penetration depth of SPRW depending on the fiber structure, the thickness and distribution of metal NPs and other sensitive layers [117].

### 5.2. Metal NPs Role and Optimization

The concentration of enzyme and the size of NPs should also be considered in the fabrication of fiber-optic biosensors. Too low concentration of enzyme and too small size (<15 nm) of metal NPs may lead to the failure of the enzyme modification [118]. In the fiber optic SPR biosensor probe, both markers, and metal NPs contribute to the high sensitivity and low LOD [119]. Regardless of selectivity, the lack of either of the two factors will greatly affect the sensing performance. The scale and distribution of metal NPs of the functional membrane exerts a significant impact on the SPR effect [120]. The distance/diameter ratio between particles becomes an effective index to reflect the coupling strength of SPRW. The smaller value results in a higher SPR intensity.

Metal NPs can be spherical, rod-shaped, and random in shape, and they will be excited to emit fluorescence with a specific wavelength depending on their morphology and size [121]. Meanwhile, the fluorescence characteristics can also be used to evaluate the morphology and distribution of metal NPs on the fiber surface. With the help of some natural plants, the synthesis of metal NPs also provides a very interesting way for the development of optical fiber SPR sensors [122]. However, the consistency of NPs is poor in some work. The dynamic dissociation adsorption process of Au NPs was used to measure the concentration of heparin [123]. Due to the stronger bond energy between the negatively charged heparin and PDDA modified surface, they replace the Au NPs on the surface of optical fiber and affect the peak wavelength of SPR resonance. In this work, the larger size of Au NPs on the surface of the optical fiber becomes much more unstable and can more sensitively sense the change of the concentration of the substance to be measured, but the structural stability of the response optical fiber probe is poor. Using the metal NPs and nanofilms with different sizes or structures, the multiple SPR resonance peaks can be obtained to realize a simultaneous analysis of different samples [124]. The seed-mediated growth technique was reported to adjust the particle size. The citrate ions were used to reduce Au^+^ ions to Au^0^ on the surface of Au NPs seeds during the Au capping process [125]. One can adjust the particle size by changing the growth time [126]. In some work, the introduction of Au NPs does not significantly improve the sensing performance of the fiber probe, which may be due to the fact that the designed structure cannot effectively change the resonance condition of the SPR effect [127]. The structure of the optical fiber, the properties of functional materials, and the morphology or structure of NPs are all factors to be included.

Metal NPs can also be modified on the surface of materials with specific structures before they are modified onto fiber structures. Graphene provides an ideal two-dimensional planar substrate and binding sites for NPs [128]. The three-dimensional functional structure was constructed by stacking multilayer 2D planar graphene [129]. The appropriate number of layers can improve the sensitivity of the sensor, while too many layers will hinder the effective action of the specimen and the optical signal. Compared with the two-dimensional functional film of Au NPs, ZnO nanowires and three-dimensional structure of Au NPs show the better SPR excitation efficiency and sensing performance, which is attributed to the fact that ZnO reduces the optical loss by capturing optical signal and improves the LSPR efficiency through electric field enhancement effect [130]. 

2-aminoethanethiol (AET) and p-mercaptophenylboronic acid (PMBA) modified Au NPs optical fiber sensor was used to measure blood glucose in urine, showing low detection limit and high selectivity [131]. For the selective detection ability of complex components, Yuan et al. adopted the molecular imprinting technique (MIPT) to construct the SPR structures with specific morphologies and used for the accurate analysis of the species and components of the analytes. On the surface of a metal NPs function layer, the synthetic recognition sites were created by MIPT using conductive polymers or micro/NPs [132]. The ratio of Ag NPs and chitosan doped in the sol determines the information conversion efficiency between the analytes and the light. The photopolymerization time and dip time of the sol will change the thickness of the functional film [133]. Moreover, the concentration of glucose in the sol of metal NPs, the liquid mixing, and the detection methods may damage the tested substance, so pretreatment of the test sample is required [134]. MIPT has been used to construct the characteristic structures in polymers, sols, and micro/NPs to realize selective recognition and concentration analysis of specific molecules [135]. The application of MIPT, SPR, and LSPR technology in optical fiber sensors can refer to the review of Gupta et al. [136]. 

The combination of LSPR of metal NPs and SPR of nano-film has also become an effective method to improve the performance of SPR optical fiber sensor [137]. It also provides more abundant research content for the structure design of the new optical fiber biochemical sensor. As the function film, PBA Au NPs were used to differentiate the RNA and DNA first, showing excellent selectivity for biomolecules based on the SPR effect of metal NPs [138]. CuS NPs also exhibit the SPR effect and are applied to exploring the LSPR biosensors [139]. In addition, the photothermal effect of metal materials has been revealed and verified by experiments, which has the potential of photothermal diagnosis and treatment. The intensity of the SPR resonance peak will change linearly because the optical power of high-order mode changes with the concentration of the analytes [140]. For the biochemical sensor probe based on the SPR effect and optical interferometer structure, the change of wavelength position or power can be obtained by demodulation. The demodulation method must refer to the changing law of spectral characteristic parameters in the process of sensor calibration. The sensing properties of the related probes are compared in Table 3.

## 6. Discussion and Perspective

The properties for metal NPs based LSPR fiber sensors have been compared with those of plasmonics devices and fiber sensors in Figure 7.

The sensitivity, response time, compact size, fabrication cost, and flexible design for these techniques have been compared according to the recent works. The LSPR fiber sensors and plasmonics devices were fabricated based on metal nanostructures, having ultra-high sensitivity compared to fiber sensors. The optical fiber supports the high transmission efficiency for the light, resulting in a faster response, which also depends on the size of the devices. LSPR fiber sensors become more compact when the palsmonics structures are prepared on the end or side surface of the optical fiber. Attributing to the focus ion beam or electron beam lithography, the high fabrication cost of plasmonics devices hinders their commercial application way. Either LSPR fiber sensors or common fiber sensors are free-standing and can be integrated with the planar devices.

The reproducible manufacturing technology of probes urgently needs a breakthrough in the future. This must refer to the production and integration methods of plasmonics devices in order to combine them with planar devices through waveguide coupling or sol packaging technology [141,142]. Corresponding sensor arrays and practical sensors are also expected to be developed, similar to the plasmonics chips [143,144]. During the design and fabrication process of metal-NPs-LSPR based optical fiber probes, the size and uniformity of metal NPs, the characteristics and modification methods of auxiliary functional materials, as well as the design and optimization of optical fiber structures should be concerned. Failure to address these issues will result in the low repeatability of such sensors and the limitation of their extensive commercial applications. New nanomaterials and nanofabrication technologies also provide the possibility for the fine design and performance optimization of such probes. As a typical candidate of the biochemical sensors, its response time and recovery effect need special attention. For example, how to improve the interaction efficiency between the LSPR effect and the molecules to be measured? Or how to quickly recover the metal NPs from the contaminated state after the measurement, so as to improve its usage lifetime and reduce the cost.

Inspired by the application of traditional optical fiber sensors, in addition to miniature biochemical sensor probes in environmental pollution, medical assistance, and biomolecular recognition, the application of LSPR sensors can also be extended to structural health monitoring of bridges, dams, and smart cities. However, the current metal NPs-based LSPR sensor is still limited to the development stage of novel biochemical probes in a laboratory environment. This is mainly due to the poor production repeatability, large differences in individual performance, unstable long-term work, and short service life. The preparation process and modification methods of metal NPs, the precise design of metal nanostructures, and the ingenious design of optical fiber structures will be important research directions to promote the development of relevant practical LSPR sensors.

## 7. Conclusions

In this paper, the development of metal-NPs-LSPR based optical fiber probes is reviewed. The fabrication methods of this kind of fiber probes include the polymer doping-dispersion of metal NPs, nano-photolithography to construct metal nanostructures, dip coating film method, electrostatic layer by layer self-assembly method, and molecular imprinting method. Based on the above technologies, the LSPR effect can also be significantly improved by changing the structure of the metal NPs functional film, laser irradiation technology for NPs size modulation, low dimensional material assisted enhancement, fluorescent material modification, and oriented sensitization of molecular markers. Metal NPs functional films have been modified on U-shaped fiber, D-typed fiber, micro/nanofiber, fiber grating, cascaded fiber structure, and polymer fiber, which are used to develop different types of probes to measure refractive index, heavy metal ions, pH value, gas, nucleic acid, and virus molecules. The simultaneous measurement of different components and concentrations is expected to be realized with the help of the different metal NPs functional films with the special excitation wavelengths and the various fiber structures working on different optical principles.

## Figures and Tables

**Figure 1 sensors-20-05155-f001:**
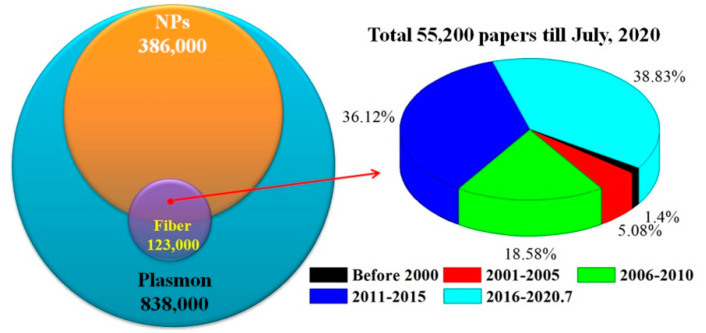
Number comparison of published paper for different key words and interval years.

**Figure 2 sensors-20-05155-f002:**
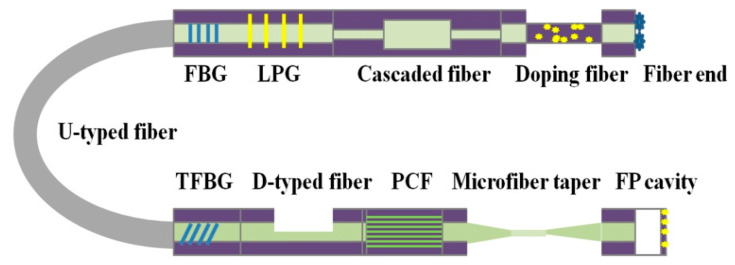
Common fiber structures for exploring LSPR fiber sensors.

**Figure 3 sensors-20-05155-f003:**
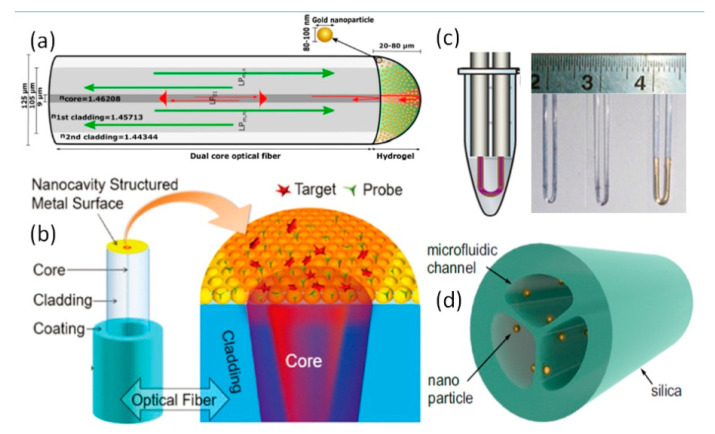
Metal NPs elaborated (**a**) fiber end using dip-coating [28]; (**b**) fiber end using self-assembly method [29]; (**c**) U-shaped fiber [40] and (**d**) air hole of PCF [41].

**Figure 4 sensors-20-05155-f004:**
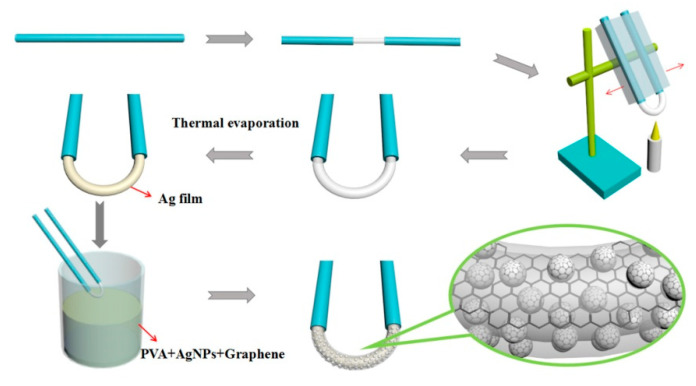
Fabrication process of U-shaped LSPR sensor probe [45].

**Figure 5 sensors-20-05155-f005:**
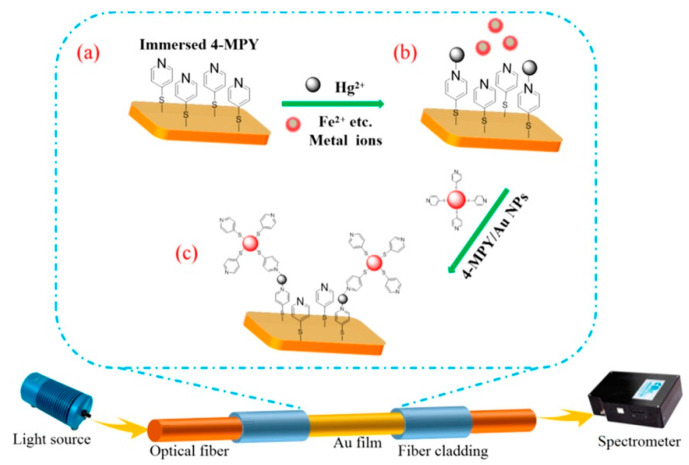
Sensing mechanism of LSPR fiber Hg^2+^ sensor. Capture of (**a**) 4-MPY and (**b**) Hg^2+^; (**c**) Selectively bind of Hg^2+^ on Au NPs surface [86].

**Figure 6 sensors-20-05155-f006:**
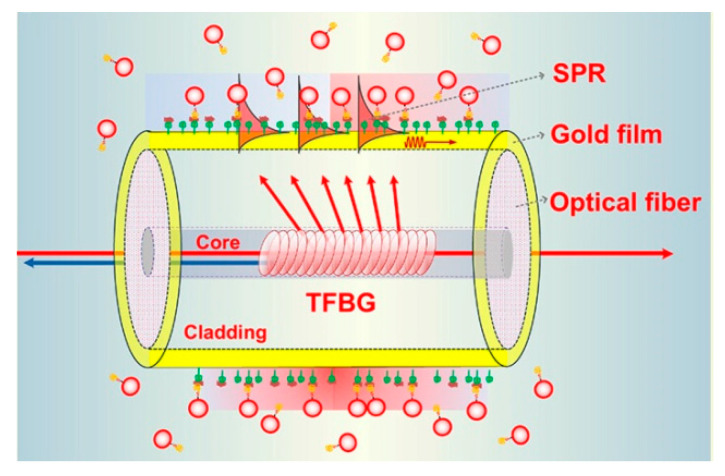
Configuration of Au NPs LSPR fiber sensor based on TFBG [106].

**Figure 7 sensors-20-05155-f007:**
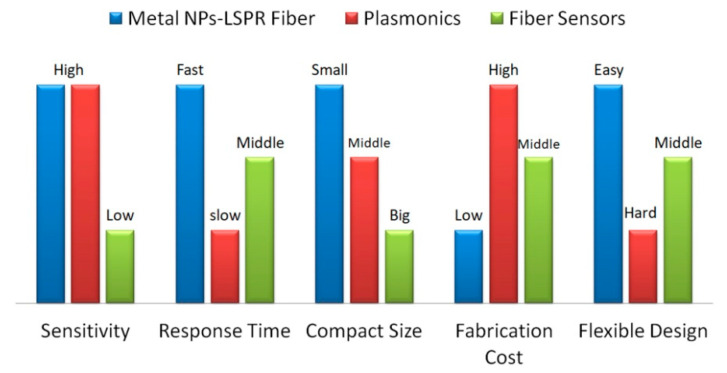
Properties comparison for metal NPs based LSPR fiber sensors, plasmonics devices and fiber sensors.

**Table 1 sensors-20-05155-t001:** Refractive index sensing performance comparison of fiber-LSPR sensors.

Fiber Structures and Elaborate Materials	Sensitivity	Working Range	Reference
U-fiber/PVA/G/AgNPs@Ag film	700.3 nm/RIU	1.330–1.3657	[45]
U-fiber/Ag NPs-Graphene	1198 nm/RIU	1.3657–1.3557	[46]
MMF/Au NPs	~987.85 nm/RIU	1.328–1.377	[47]
MMF/PDA-Ag NPs-PDA	961 nm/RIU	1.33–1.40	[48]
MMF/Au NRs	75.69 dB/RIU	1.33–1.408	[49]
Microfiber taper/Au-Ag nanoprisms	900 nm/RIU	1.333–1.383	[50]
Side-polished PCF/Ag NR	8600 nm/RIU	1.33–1.40	[51]
MMF-SMF-MMF/Au NPs	765 nm/RIU	1.333–1.365	[52]
LPFG/Au NPs	~3928 nm/RIU	1.3333–1.3428	[53]
Core angle SMF-MMF/Au NPs-Au film	5140nm/RIU	1.32–1.37	[54]
U-fiber/Spherical Ag NPs	342.7 nm/RIU	1.3317–1.3640	[55]
U-fiber/Triangular Ag NPs	1116.8 nm/RIU		
MMF/Au Multibranched NPs	3164 nm/RIU	1.333 to 1.393	[56]
MMF/Au NPs	349.1 nm/RIU	1.333–1.403	[57]
D fiber/Au NPs-Au film	3074 nm/RIU	1.3332–1.3710	[58]
MMF-SMF-MMF/Graphene-Au-Au@Ag NPs-PDMS	1591 nm/RIU	1.3330–1.4005	[59]
MMF end/MWCNT-Pt NPs	5923 nm/RIU	1.3385–1.3585	[60]

**Table 2 sensors-20-05155-t002:** Chemical sensing performance comparison for fiber-LSPR sensors.

Fiber Structures and Elaborate Materials	Measurement Target	Sensitivity/Resolution	Working Range	Reference
Fiber nanotaper/Au NPs	R6G	0.1 μM	0.1–100 μM	[64]
	R6G	1 μM/L	0.1–100 μM	[65]
U-fiber/Au NPs	Gasoline	0.1198 mV/ppm	0–153.38 ppm	[66]
		0.08244 mV/ppm	0–144.09 ppm	
U-fiber/Au NPs-WS2 film	Ethanol	0.65/%	10–80%	[67]
	NaCl	1.5/%	5–25%	
Fiber tip/Ag NPs-4 MPY	PH	0–1 cm^−1^/PHU	5.01–9.10	[68]
NCF/Au NPs-PHPH	PH	~25 counts/PHU	1–12	[69]
Dual fiber tips/Au NPs	Rh B	10 ppm (LOD)	--	[71]
Zein nanofiber/Au NPs	Rh 6G	100 μM	100 μM–1 mM	[72]
MMF tip/Au NPs	Rh 6G	--	50 μM–1 mM	[73]
NCF/CNT-Cu NPs	Nitrate	80.62 × 10^6^ nm/M	10^−^^6^ M–5 × 10^−^^3^ M	[74]
U-fiber/Glucose-Ag NPs	Hg^2+^	2 ppb	0–200 ppb	[77]
MMF/PAA-capped AuNPs	Hg^2+^	0.7 ppb	1–20 ppb	[78]
NCF/PVA-Au NPs	Hg^2+^	1 μM	0–25 μM	[79]
LPG/PE-Au NPs	Hg^2+^	0.5 ppm	0.5–10 ppm	[80]
NCF/ternary system-Cu@Ag NPs	Hg^2+^	0.01 μM	0.01–1000 μM	[81]
U-fiber/Oxalic acid-Au NPs	Pb^2+^	1.75 ppb	1–20 ppb	[82]
Fiber end/DNA-Au NPs	Hg^2+^	0.7 nM	1–50 nM	[83]
U fiber/BSA-Chitosan-Au NPs	Hg^2+^	0.1 ppb	0.1–540 ppb	[84]
U-fiber/Au NPs-PSS-PAH-E.coli	Hg^2+^/Cd^2+^	0.5 ppb	2–2000 ppb	[85]
NCF/Au NPs-4 MPY	Hg^2+^	8 nM	8–100 nM	[86]
NCF/Au NPs-MUA	Pb^2+^	800 μM (65 ppm)	10–100mM	[88]
Planar waveguide/ chitosan-Au/Ag NPs	H_2_S	5 ppm	5–300 ppm	[90]
		0.1 ppm	0.1–100 ppm	
U-fiber/Au(Ag) NPs	methanol	~26.46 (2.12) ppm	0–228.65 ppm	[91]
	acetone	~1.00 (0.43) ppm	0–202.72 ppm	
	ethanol	~3.53 (3.79) ppm	0–304.26 ppm	
	propanol	~3.81 (11.09) ppm	0–148.89 ppm	
Microfiber taper/Pt NPs-GO	NH_3_	10.2 pm/ppm	0–120 ppm	[92]
Microfiber/Pd NPs	H_2_	52 pm/ppm	0.2–1 × 10^4^ ppm	[93]
Ag-Pt NPs-ZnO	Ethanol/acetone	0.5 ppm	0–100 ppm	[94]
U-fiber/Pt-WO3 nanosheets	H_2_	43.5 ppm	0–1.6 × 10^4^ ppm	[95]

4-mercaptopyridine (4-MPY); Phenolphthalein (PHPH); Carbon Nanotubes (CNT); Poly Acrylic acid (PAA); Polyethaline (PE); Non-core Fiber (NCF); Oxalic Acid (OA); Bovine Serum Albumin (BSA); Poly(sodium-p-styrenesulfonate) (PSS); Poly(allylamine hydrochloride (PAH); 1, 1-Mercaptoundecanoic acid (MUA).

**Table 3 sensors-20-05155-t003:** Performance comparison of LSPR biosensors based on different structures and materials.

Fiber Structure/Materials	Target	Limit of Detection	Sensitivity	Working Range	Reference
Flattened fiber/Ag NPs	Glucose	4.42 mg/dL	--	0–500 mg/dL	[96]
U-fiber/Au NPs-GOD	Glucose	0.16 mg/dL	2.899 nm/%	0.1–0.5%	[98]
PMMA U-fiber/Tyr-AgNPs	Dopamine	0.16 μM		0–50 μM	[99]
Microfiber Taper/SiO_2_-Au NPs	SV	271 pM	--	2.5 nM–1.33 μM	[100]
Microfiber taper/Au NPs	Cholesterol	53.1 nM	0.125%/mM	10 nM–1 μM	[101]
Microfiber taper/Au NPs	UA	175.89 μM	0.0131 nm/μM	10–800 μM	[102]
	BSA	0.3263 gm/dL	~25 μA/mM	0.05–0.2 mM	[59]
HCF Tip/Ag NSs-NRs	MB	10 fM	--	--	[104]
	Melamine	100 nM			
LPG/Au NPs-Cys	Glyphosate	0.02 μM	3.5 nm/μM	0.5–100 μM	[105]
TFBG/Au NPs-film	Thrombin	1 nM	3.21 × 10^7^ dB/M	1–33.75 nM	[106]
PCF/Au NPs-film	Human IgG	37 ng/mL	~0.54 pm/(ng/mL)	1–30 μg/mL	[109]
Microfiber taper/Au NPs	SV	1 pg/mL		0 pg–1 ng/mL	[110]
HCF/Ag NPs	Cholesterol	25.5 nM	16.149 nm/μM	50 nM–1 μM	[111]
MMF/Au-Ag NPs-GO	L-Cysteine	126.6 μM	0.0012 nm/μM	50 μM–1 mM	[112]
Microfiber taper/Au NPs	AA	51.94 μM	8.3 nm/mM	10 μM–1 mM	[113]
MMF End/Au NPs	PSA	124 fg/mL	--	1 pg–10μg/mL	[114]
MMF/SMF End/Au NPs		0.1 pg/mL	--	0.1–100 pg/mL	[115]
LPG/SiO_2_-Au NPs	IgM	15 pg/mm^2^	11 nm/(ng/mm^2^)	15 μg–1mg/mL	[116]
LPG/SiO_2_-Au NPs	SV	0.86 pg/mm^2^	3.88/(ng/mm^2^)	1.25 nM–2.7 μM	[117]
NCF/Ag NPs	TG	0.016 mM	28.5 nm/mM	0–7 nM	[118]
NCF/Dios-Ag NPs	Cysteine	0.0077 μM		0–100 μM	[119]
MMF/PS-b-P4VP-Au NPs	Human IgG	0.3nM	38 pm/(ng/cm^2^)	6.7–66.7 nM	[120]
NCF/Au NRs	OTA	12.0 pM	--	10 pM–100 nM,	[121]
MMF/PVA-Ag NPs-GO	Dopamine	0.2 μM	--	0.2–80 μM	[122]
NCF/PDDA-Au NPs	Heparin	0.0257ng/mL	1 nm/(ng/mL)	0.1ng-1μg/mL	[123]
MMF-PCF/Ag-Au film-NPs-GO	Human IgG	15 ng/mL	~13.6 μm/RIU	1–40 μg/mL	[124]
MMF end/Au NPs	Tg	0.19 pg/mL	--	1 pg–10 ng/mL	[125]
Microfiber Taper/GO-Au NPs	UA	259 μM	8.9 pm/μM	10–800 μM	[127]
U-MMF/Graphene-Au NPs	DNA	0.1 nM	1.25 μm/RIU	0.1–100nM	[129]
Flattened fiber/Au NPs or ZnO NWs-Au NPs	PSA	2.06 pg/mL	35 V/RIU	0.01 pg–1 ng/mL	[130]
	PSA	0.51 pg/mL	60 V/RIU		
NCF/Au NPs	Glucose	80 nM	~1.44 nm/nM	0.01–30 mM	[131]
NCF/Ag NPs	ERY	1.62 nM	205 nm/μM	0–100 μM	[132]
MMF/Ag NPs	TCA	10 μM	0.587 nm/μM	40–100 μM	[133]
NCF/Au NPs-SnO_2_	Dopamine	0.031 μM,	--	0–100 μM	[135]
NCF/Ag NPs-GO	Cholesterol	1.131 mM	5.068 nm/mM	0–10 mM	[137]
NCF/PBA-Au NPs	MicroRNA	0.27 pM	--	10 pM–10 μM	[138]
U-Microfiber/GO-CuS NPs	MicroRNA	0.0156 aM	0.62 nm/lgM	0.1 aM–10 pM	[139]
MMF/Au NPs	Taurine	53 μM	0.0190 AU/mM	0–1 mM	[140]

Streptavidin (SV); Uric Acid (UA); Methylene Blue (MB); Ascorbic Acid (AA); Immunoglobulin M (IgM); Phenylboronic Acid (PBA); Glucose Oxidase (GOD); Thyroglobulin (Tg); Trichloroacetic Acid (TCA); Erythromycin (ERY); Prostate-Specific Antigen (PSA); Triacylglycerides (TG); Ochratoxin A (OTA).

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
