# Peer review of "Preparation and Application of Metal Nanoparticals Elaborated Fiber Sensors"

_sensors, 2020, doi:10.3390/s20185155_

Round 1

Reviewer 1 Report

In this review paper, Li et al, summarized the fiber sensors elaborated metal nanoparticles. They reviewed the fiber-type sensors to detect physical parameters, environmental parameters and bio-related molecules. They listed various kinds of fiber sensors with nanoparticles. The mechanistic action, in particular. The detection methods are missing, therefore, I suggest that the authors add the separated paragraph to explain why the nanoparticle embedding into fiber improves the signals such as OD and Raman emission for the publication to the Sensors.

Author Response

Reply: A new paragraph was included to explain the mechanism of the enhancement for OD and Raman emission.

Revision: See the modification on Page 13, lines 439-447 and references [108] and [109]:

"Compared with the LRSPR sensor, the LOD and surface enhancement effects of the LSPR sensor are more obvious. Noble metal NPs provide a larger contact area, and achieve efficient information exchange between the metal NPs and the analyte. More importantly, the NPs surface generates a higher concentration of electromagnetic field and the limitations of single-point amplification effect, which can effectively enhance the detection efficiency of a biological analyte in nanometer size or spacing [108]. In the detection process, the excitation photons meeting the coupling frequency resonate with the surface electrons of the metal NP, thereby generating a stronger and concentrated surface-local evanescent field, and realizing the single-molecule nanoprobe having a very low LOD [109]."

[108] Chen, H. M.; Zhao, L.; Chen, D. Q.; Hu, W. H., Stabilization of gold nanoparticles on glass surface with polydopamine thin film for reliable LSPR sensing. J Colloid Interf Sci 2015, 460, 258-263.

[109] Mayer, K. M.; Hafner, J. H., Localized Surface Plasmon Resonance Sensors. Chem Rev 2011, 111, (6), 3828-3857.

Reviewer 2 Report

In this manuscript, the development of metal-NPs-LSPR based optical fiber probes is reviewed. The fabrication methods of this kind of fiber probes include the polymer doping-dispersion of metal NPs, nano-photolithography to construct metal nanostructures, dip coating film method, electrostatic layer by layer self-assembly method and molecular imprinting method. Overall, this review is timely and organized well. I sugguest it can be accepted after a minor revision to solve the following concerns.

  • For clear illustration, it is suggested that the authors add a picture to embed the obvious advantages (maybe featured by the high sensitivity, compact device size, fast response time, low fabrication cost, and etc.) of metal nanoparticals elaborated fiber sensors compared with other sensing technologies (e.g., fiber sensing and surface plasmonic sensing).
  • The authors mainly focus on the devices’level design and performance. In order to show the great application potentials. It is suggested that systems’level applications in bridge, tunneling, smart city, health, medical treatment or other occasion are to be reviewed.
  • The main challenge and future developing trend of this technology are suggested to be discussed in detail?
  • Some relevant references may be helpful enrich the manuscript, e.g., Refractive Index Sensing Research on Multi-Fano-Based Plasmonic MDM Resonant System With Water-Based Dielectric,IEEE Journal of Quantum Electronics , 2020 , 56 (3) :1-7; On-chip readout plasmonic mid-IR gas sensor. Opto-Electron Adv 3, 190040 (2020); Multiple Fano Resonances Based on End-Coupled Semi-Ring Rectangular Resonator, IEEE Photonics Journal , 2019 , 11 (4) :1-8; Ultrasensitive skin-like wearable optical sensors based on glass micro/nanofibers. Opto-Electron Adv 3, 190022 (2020)

Author Response

Reviewer 2:

In this manuscript, the development of metal-NPs-LSPR based optical fiber probes is reviewed. The fabrication methods of this kind of fiber probes include the polymer doping-dispersion of metal NPs, nano-photolithography to construct metal nanostructures, dip coating film method, electrostatic layer by layer self-assembly method and molecular imprinting method. Overall, this review is timely and organized well. I suggest it can be accepted after a minor revision to solve the following concerns.

  1. For clear illustration, it is suggested that the authors add a picture to embed the obvious advantages (maybe featured by the high sensitivity, compact device size, fast response time, low fabrication cost, and etc.) of metal nanoparticals elaborated fiber sensors compared with other sensing technologies (e.g., fiber sensing and surface plasmonic sensing).

Reply: The characteristics of the metal-NPs-LSPR-fiber sensors have been compared with those of fiber and plasmonics sensors.

Revision: See the modification on Page 15, lines 540-553 and Figure 7:

"The properties for metal NPs based LSPR fiber sensors have been compared with those of plasmonics devices and fiber sensors in Figure 7.

Figure 7. Properties comparison for metal NPs based LSPR fiber sensors, plasmonics devices and fiber sensors.

The sensitivity, response time, compact size, fabrication cost and flexible design for these techniques have been compared according to the recent works. The LSPR fiber sensors and plasmonics devices were fabricated based on metal nanostructures, having the ultra-high sensitivity compared to fiber sensors. The optical fiber supports the high transmission efficiency for the light, resulting in the faster response, which also depends on the size of the devices. LSPR fiber sensors become more compact when the palsmonics structures are prepared on the end or side surface of optical fiber. Attributing to the focus ion beam or electron beam lithography, the high fabrication cost of plasmonics devices hinders their commercial application way. Either LSPR fiber sensors or common fiber sensors are free standing and can be integrated with the planar devices."

  1. The authors mainly focus on the devices’level design and performance. In order to show the great application potentials. It is suggested that systems’level applications in bridge, tunneling, smart city, health, medical treatment or other occasion are to be reviewed.

Reply: At this stage, the applications of the metal NPs based fiber sensors are limited for the fabrication technique or the few study on their integration with other fiber systems. The systems' level applications have been discussed.

Revision: See the modification on Page 16, lines 569-578:

"Inspired by the application of traditional optical fiber sensors, in addition to miniature biochemical sensor probes in environmental pollution, medical assistance and biomolecular recognition, the application of LSPR sensors can also be extended to structural health monitoring of bridges, dams and smart cities. However, the current metal NPs-based LSPR sensor is still limited to the development stage of novel biochemical probes in laboratory environment. This is mainly due to the poor production repeatability, large differences in individual performance, unstable long-term work and short service life. The preparation process and modification methods of metal NPs, the precise design of metal nanostructures, and the ingenious design of optical fiber structures will be important research directions to promote the development of relevant practical LSPR sensors."

  1. The main challenge and future developing trend of this technology are suggested to be discussed in detail?

Reply: We have discussed the challenge and future development trend of this technology.

Revision: See the modification on Pages 15-16, lines 554-568:

"The reproducible manufacturing technology of probes urgently needs a breakthrough in the future. This must refer to the production and integration methods of plasmonics devices in order to combine them with planar devices through waveguide coupling or sol packaging technology [142, 143]. Corresponding sensor arrays and practical sensors are also expected to be developed, similar to the plasmonics chips [144, 145]. During the design and fabrication process of metal-NPs-LSPR based optical fiber probes, the size and uniformity of metal NPs, the characteristics and modification methods of auxiliary functional materials, as well as the design and optimization of optical fiber structures should be concerned. Failure to address these issues will result in the low repeatability of such sensors and the limitation of their extensive commercial applications. New nanomaterials and nanofabrication technologies also provide the possibility for the fine design and performance optimization of such probes. As a typical candidate of the biochemical sensors, its response time and recovery effect need the special attention. For example, how to improve the interaction efficiency between LSPR effect and the molecules to be measured? Or how to quickly recover the metal NPs from the contaminated state after the measurement, so as to improve its usage life time and reduce the cost."

  1. Some relevant references may be helpful enrich the manuscript, e.g., Refractive Index Sensing Research on Multi-Fano-Based Plasmonic MDM Resonant System With Water-Based Dielectric,IEEE Journal of Quantum Electronics , 2020 , 56 (3) :1-7; On-chip readout plasmonic mid-IR gas sensor. Opto-Electron Adv 3, 190040 (2020); Multiple Fano Resonances Based on End-Coupled Semi-Ring Rectangular Resonator, IEEE Photonics Journal , 2019 , 11 (4) :1-8; Ultrasensitive skin-like wearable optical sensors based on glass micro/nanofibers. Opto-Electron Adv 3, 190022 (2020)

Reply: The suggested papers have been cited and discussed in the revised manuscript.

Revision: See the modification on Page , lines and References [142], [143], [144], [145]:

"The reproducible manufacturing technology of probes urgently needs a breakthrough in the future. This must refer to the production and integration methods of plasmonics devices in order to combine them with planar devices through waveguide coupling or sol packaging technology [142, 143]. Corresponding sensor arrays and practical sensors are also expected to be developed, similar to the plasmonics chips [144, 145]. "

  1. Fang, Y. H.; Wen, K. H.; Li, Z. F.; Wu, B. Y.; Chen, L.; Zhou, J. Y.; Zhou, D. Y., Multiple Fano Resonances Based on End-Coupled Semi-Ring Rectangular Resonator. Ieee Photonics J 2019, 11, (4), 1-8.
  2. Zhang, L.; Pan, J.; Zhang, Z.; Wu, H.; Yao, N.; Cai, D. W.; Xu, Y. X.; Zhang, J.; Sun, G. F.; Wang, L. Q.; Geng, W. D.; Jin, W. G.; Fang, W.; Di, D. W.; Tong, L. M., Ultrasensitive skin-like wearable optical sensors based on glass micro/nanofibers. Opto-Electron Adv 2020, 3, (3), 190022.
  3. Li, Z. F.; Wen, K. H.; Fang, Y. H.; Guo, Z. C., Refractive Index Sensing Research on Multi-Fano-Based Plasmonic MDM Resonant System With Water-Based Dielectric. Ieee J Quantum Elect 2020, 56, (3), 1-7.
  4. Chen, Q.; Liang, L.; Zheng, Q. L.; Zhang, Y. X.; Wen, L., On-chip readout plasmonic mid-IR gas sensor. Opto-Electron Adv 2020, 3, (7), 190040.